

# ESD Ideas: The Peclet number is a centerstone of the orbital and millennial Pleistocene variability

*Mikhail Y. Verbitsky[1,2] and Michel Crucifix[2]*

[1]Gen5 Group, LLC, Newton, MA, USA
[2]UCLouvain, Earth and Life Institute, Louvain-la-Neuve, Belgium

5                 Correspondence: Mikhaïl Verbitsky (verbitskys@gmail.com)

**Abstract.** We demonstrate here that a single physical phenomenon, specifically, a naturally changing balance between intensities of temperature advection and diffusion in the viscous ice media, may influence the entire spectrum of the Pleistocene variability from orbital to millennial time-scales.

**Introduction.** About 1 million years ago, the dynamics of glacial-interglacial cycles experienced a major change, transitioning from the predominant 40-kyr periodicity to approximately 100-kyr variability (e.g., Ruddiman et al., 1986; Lisiecki and Raymo, 2005). This change of the glaciation rhythmicity is called the middle-Pleistocene transition (MPT). On the other hand, Hodell and Channell (2016) presented the analysis of 3.2 Myr records of stable isotopes and sediment physical properties, and observed coherence

between millennial and orbital-scale variabilities: after the MPT, millennial events are more frequent and more intense than before. In our previous work (Verbitsky et al., 2019), we have shown that millennial variability may penetrate upscale and influence the slower pace dynamics of glacial-interglacial cycles. Here, we will demonstrate that the same ice sheet non-linearity that is responsible for the penetration of the millennial oscillations into the orbital domain may also contribute — without invoking any additional

physics — to the millennial variability and explain its coherence with orbital-scale dynamics. To underline the significance of this proposal, we briefly summarize the current understanding of millennial variability.

    Classically, millennial variability during the glacial periods is explained by reference to two families of mechanisms acting in concert. The first one relates to iceberg calving, which occurs when ice sheets have reached the ocean margins and developed ice-shelves. Episodes of intense ice-rafting generate a halocline

and reduce ocean ventilation (the *Heinrich* events); on the other hand, changes in ocean temperature and sea-level influence the stability of ice shelves and may pre-condition calving events. The second family of mechanisms involves the dynamics of ocean circulation, deep-water mixing and sea-ice formation at high latitudes. The ocean circulation effectively causes horizontal and vertical transports of buoyancy, which have a role in maintaining the circulation itself. Because of the non-linear character of this feedback, the

geographical structure of convection in the North Atlantic realm can shift very rapidly (within years) between different configurations. There is converging evidence that the *Dansgaard-Oeschger* events identified in Greenland correspond to such circulation changes. Millennial variability associated with all these mechanisms is most likely to occur during cold periods (ice sheets must be large enough) but not too cold (sea-ice must not be locked into a deep freeze): a so-called "sweet-spot" (Mitsui et al., 2018; Pedro et

al., 2018; Li and Born, 2019) during which iceberg calving and ocean-circulation changes can entertain a rich pattern of oscillations (Schulz et al., 2002). The mechanism that we suggest here is not subject to the same sweet spot, and can actually explain a pervasive variability during both glacial *and* interglacial periods.

    **Methods.** We consider the non-linear dynamical model of the global climate system presented in

(Verbitsky et al, 2018). It is derived from the scaled conservation equations of the non-Newtonian ice flow, and combined with a linear feedback equation of the climate temperature:

$$\frac{dS}{dt} = \frac{4}{5}\zeta^{-1}S^{3/4}(a - \varepsilon F_S - \kappa\omega - c\theta) \tag{1}$$

$$\frac{d\theta}{dt} = \zeta^{-1}S^{-1/4}(a - \varepsilon F_S - \kappa\omega)\{\alpha\omega + \beta[S - S_0] - \theta\} \tag{2}$$

$$\frac{d\omega}{dt} = \gamma_1 - \gamma_2[S - S_0] - \gamma_3\omega \tag{3}$$


Equations (1) - (3) describe dynamical properties of a large continental (e.g., Laurentide) ice sheet. Here $S$ (m$^2$) is the glaciation area, $\theta$ ($^o$C) is the basal ice sheet temperature, and $\omega$ ($^o$C) is the global climate temperature. The profile factor $\zeta$ (m$^{1/2}$) is determined by ice viscosity, density, and acceleration of gravity; $a$ (m/s) is snow precipitation rate; $F_S$ is normalized mid-July insolation at 65°N (Berger and Loutre, 1991);

$\varepsilon$ (m/s) is the amplitude of $F_S$; $\kappa$ (m s$^{-1}$ $^o$C$^{-1}$) and $c$ (m s$^{-1}$ $^o$C$^{-1}$) define ice mass balance sensitivity to $\omega$ and $\theta$; adimensional coefficient $\alpha$ is the basal temperature sensitivity to $\omega$ changes, $\beta$ ($^o$C /m$^2$) defines basal temperature sensitivity to the changes of the ice sheet area $S$, $S_0$ (m$^2$) is a reference glaciation area; $\gamma_1$ ($^o$C/s), $\gamma_2$ ($^o$C m$^{-2}$ s$^{-1}$) and $\gamma_3$ (s$^{-1}$) define $\omega$ evolution.





We now supplement this system with equations (4) and (5) that describe the thermodynamical properties of a smaller-size (e.g., Greenland) ice sheet.

$$\frac{dS_G}{dt} = \frac{4}{5}\zeta^{-1}S_G^{3/4}(a' - \varepsilon'F_S - \kappa'\omega - c'\theta_G) \tag{4}$$

$$\frac{d\theta_G}{dt} = \zeta^{-1}S_G^{-1/4}(a' - \varepsilon'F_S - \kappa'\omega)\{\alpha'\omega + \beta'(S_G - S'_0) - \theta_G\} \tag{5}$$

Equations (4) and (5) are identical to equations (1) and (2). Here $S_G$ (m$^2$) and $\theta_G$ ($^{\circ}$C) are the area and the basal temperature of the Greenland ice sheet; all other parameters have the same dimensions and meaning as the corresponding parameters in the equations (1) and (2). Some (but not all) of them may have different numerical values, and for this reason we mark them with an apostrophe. The area of the Greenland ice sheet is limited by the size of the island and its variations have limited impacts on the global climate. Therefore, for reasoning on scaling relationships, we can neglect its contribution to the $\omega$-evolution and leave equation (3) unchanged. Hence, the Greenland ice sheet model may be reduced to a system formed by equations (4) and (5), forced by the astronomical forcing and the global climate temperature $\omega$. The dynamical system formed by the equations (1) – (5) is represented graphically on Fig. 1(a). Without external forcing, the evolution of $S_G$ and $\theta_G$ is fully determined by 5 parameters (namely $a'$, $\varsigma$, $S'_0$, $\beta'$, and $c'$). Some combinations of these parameters generate a relaxation-oscillation behavior, and as we show now, it is reasonably straightforward to estimate the period $P$ of this oscillation. Indeed, if we take parameters $a'$, $S'_0$, and $\beta'$, as parameters with independent dimensions, and consider parameters $\varsigma$ and $S'_0$ as constant, then, using the generalized $\pi$-theorem (Sonin, 2004), the time scale $P$ obeys:

$$P = (a')^{-1}S'^{1/2}_0\Psi(\Pi_1) \tag{6}$$

Here $\Pi_1 = a'/(\beta'c'S'_0)$. It can be determined experimentally that period $P$ is smaller when $\Pi_1$ is smaller. Following Verbitsky and Chalikov (1986), we now introduce the Peclet number as $Pe = \hat{a}H/k$, where $\hat{a}$ is a characteristic mass influx, i.e., accumulation minus ablation, $H$ is ice thickness, and $k$ is temperature diffusivity. It measures the balance between temperature advection and diffusion. Since the Greenland ice sheet is relatively thin, its Peclet number is smaller than that of a big ice sheet, and therefore the effect of the geothermal heat flux on the basal temperature is, in relative terms, stronger. Verbitsky et al (2018) showed that parameter $\beta$, which determines the basal temperature response to the changes of the ice sheet size, emerges as a delicate balance between vertical advection, internal friction, and geothermal heat flux, and it is proportional to $Pe^{-1/2}$. Thus, the relatively small size of the Greenland ice sheet implies a higher $\beta'$ and, according to the equation (6), its relaxation oscillations have higher frequency than, say, the Laurentide ice sheet. In fact, numerical experiments with equations (4) - (5) and with a reasonable set of parameters produce millennial-range relaxation oscillations of the Greenland ice sheet.

With these considerations in hand, let us foresee the consequences of the MPT. As the Laurentide ice sheet reached gradually increasing footprints on the large American continent, its Peclet number increased, implying that temperature advection became the dominant process in the moving ice media. Consequently, $\beta$ becomes smaller. We previously showed (Verbitsky et al, 2018, Verbitsky and Crucifix, 2020) that the dynamics of the system described by equations (1) - (3) is largely determined by a $V$-number measuring a balance between positive and negative model feedbacks:

$$V = \frac{1}{\beta}\left(\alpha + \frac{\kappa}{c}\right)\left(\frac{\gamma_2}{\gamma_3} - \frac{\gamma_1}{S_0\gamma_3}\right) \tag{7}$$

Thus, high values of the parameter $\beta$ ($V\sim 0$) imply a weak positive feedback in the system, and, conversely, low values of the parameter $\beta$ ($V\sim 1$) imply a strong positive feedback. As it has been previously shown (Verbitsky et al, 2018), when the orbital forcing is large enough compared to the average ice accumulation (high enough $\varepsilon/a$ ratio), an increase in the $V$-number generates a period-doubling bifurcation, that is, an escape from the 40-kyr oscillation regime towards longer glacial-interglacial cycles. The increased period of the system response to the astronomical forcing also allows for the glaciation area $S$ to become larger, meaning that the amplitude of the oscillation is larger after the MPT than before. This amplitude increases may be understood as a consequence of a scale-invariance property (Verbitsky and Crucifix, 2020). In this





case, we see that the natural evolution of the Peclet number generates higher-amplitude oscillations of the larger ice sheets (i.e., Laurentide ice sheet), along with longer glacial-interglacial cycles. On the other hand, over Greenland, the Peclet number cannot grow. It remains bounded by a small value which allows

millennial relaxation oscillations. Yet, the large post-MPT oscillations of the Laurentide ice sheet may excite more vigorously the millennial oscillations of Greenland. The idea of the present contribution is therefore the following: *the Peclet number is a key quantity for explaining the joint emergence of large-amplitude oscillations of the bigger ice sheets, along with the increase in the occurrence of the millennial oscillations of the smaller-size ice sheets.*

To test our theoretical perspective, we reproduce the Pleistocene glaciation history, solving system (1) – (5) jointly, thus calculating a coevolution of the Laurentide and Greenland ice sheets. For the Laurentide ice sheet, we change linearly the $\varepsilon/a$ ratio over the last 3 million years from $\varepsilon/a = 0.3$ to $\varepsilon/a = 1.7$ and change the reference glaciation area $S_0$ from $S_0 = 2\ 10^6\ km^2$ to $S_0 = 12\ 10^6\ km^2$ thus invoking a hypothetical trend in processes that control long-term $CO_2$ levels. The $\varepsilon/a$ ratio is reduced by changing the $a$-component

only, following the assumption that the Pleistocene cooling trend goes along with a decrease in the snow precipitation rate. The parameter $\beta$ is modified as being proportional to $Pe^{-1/2} \sim H^{-1/2} \sim S_0^{-1/8}$ (Since $H \sim S^{1/4}$, Verbitsky et al, 2018), changing from $\beta = 2.4\ ^oC\ 10^{-6}\ km^{-2}$ to $\beta = 1.9\ ^oC\ 10^{-6}\ km^{-2}$. To account for the small effect of Greenland changes on climate, which we have neglected for our scaling reasoning, we add a term $-\gamma_2 S_G$ into the right side of the equation (3). For the Greenland ice sheet, we assume that the reference

glaciation area depends on climate temperature, i.e., $S'_0 \sim -\omega$, $\beta' \sim S'^{-1/8}_0$, and the ratio $\Pi_1$ is an order of magnitude less than the corresponding ratio of the Laurentide ice sheet. In Fig. 1(b) we present the results as a wavelet spectrum over the past 3 million years for the Greenland glaciation area $S_G$. We can see that though the Greenland ice sheet itself has a limited influence on the global climate temperature ($\gamma_2 S_G \ll \gamma_2 S$), the changes of Greenland's geometry reflect all major events of the Pleistocene history – a transition

from the double-precession 40-kyr variability to increased amplitudes of double-obliquity oscillations. The global climate temperature $\omega$, driven by the Laurentide ice sheet (taken here as a proxy for all the large ice sheets of the Northern Hemisphere), shifts the equilibrium state of the Greenland ice sheet, and the latter responds with millennial-period oscillatory adjustments. Such shifts are more prominent in the late Pleistocene and therefore, consistently with Hodell and Channell (2016), the millennial events of the late

Pleistocene are more frequent and more intense. The amplitude of $S_G$ millennial variability is $\sim 0.1\ 10^6\ km^2$, corresponding to $\sim 0.15\ 10^6\ km^3$ of ice volume or $\sim 0.4$ m of the sea level change.

**Discussion.** Until recently, the time-scale separation approach (Saltzman, 1990) has been implicitly or explicitly used to considering separately glacial-interglacial cycles and millennial variability. Accordingly, it is generally accepted that the spectrum of Pleistocene variability in the orbital domain is defined by

continental ice sheets, while the millennial part of the spectrum has been exclusively attributed to the ocean-atmosphere interactions. We first challenged this approach by demonstrating that ice-sheet non-linearity provides an avenue for millennial variability to propagate upscale, and influence ice-age dynamics (Verbitsky et al., 2019). Here, we show that the same non-linear ice-flow dynamics can also affect the millennial part of the spectrum.

From equation (7), it appears that different mechanisms would explain an increase in the $V$-number throughout the Pleistocene. Therefore, several scenarios can be invoked as the origin of the MPT. For example, a steady decrease in the background climate temperature ($S_0$, $\gamma_1$), an increase in climate sensitivity to the ice volume ($\gamma_2$), a change in the sensitivity of ice mass balance and its temperature to global climate temperature ($\kappa$ and $\alpha$), or even a decrease in the intensity of ice sliding ($c$). At the same time, all these

mechanisms produce the changes in the Peclet number which we have described. For this reason, we claim that *the Peclet number, changing in concert with the glaciation size, is the key similarity parameter connecting the millennial and orbital Pleistocene variations.* We find it remarkable that the same physical phenomenon can explain all major events of the Pleistocene in the orbital domain if the evolution of an ice sheet is not restricted, and in the millennial domain when the size of the ice sheet is fixed.


**Author contributions:** MYV conceived the research and developed the formalism. MYV and MC contributed equally in writing the paper.

**Competing interests:** The authors declare that they have no conflict of interest.

**Financial support:** Michel Crucifix is funded as Research Director with the Belgian National Fund of
Scientific Research.





**(a)**

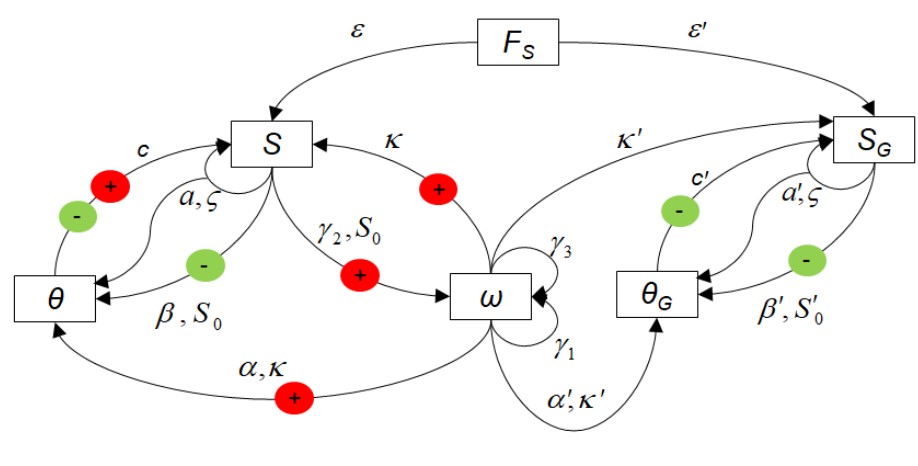

Laurentide ice sheet                    Greenland ice sheet

**(b)**

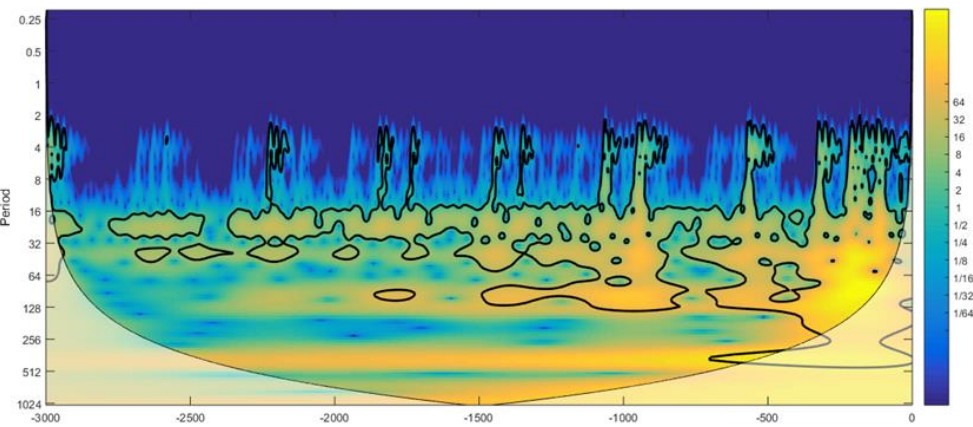

**Fig. 1: (a)** the dynamical system (1) – (5). Red circles mark positive feedback loops and green circles mark
negative feedback loops. **(b)** Evolution of wavelet spectra over the past 3 million years for the Greenland
glaciation area $S_G$ ($10^6$ km$^2$). The color scale shows the continuous Morlet wavelet amplitude, the thick line
indicates the peaks with 95 % confidence, and the shaded area indicates the cone of influence for wavelet
transform.



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
