# Peer review of "Mikhail Y. Verbitsky1,2 and Michel Crucifix2"

_Earth System Dynamics, 2020_

## Referee Comment (RC1) · Anonymous Referee #1 · 18 Oct 2020

Review of "ESD Ideas: The Peclet number is a centerstone of the orbital and millennial Pleistocene variability" by Verbitsky and Crucifix

Recommendation: Minor revisions

This short paper presents a plausible mechanism how ice sheet dynamics can affect Pleistocene variability on orbital and millennial time scales. This is a nice contribution to this important topic. The manuscript is well written and should be considered for publication in ESD.

I have only one question. In your model Eqs. (1)-(5) you represent two ice sheets (Laurentide and Greenland) which are independent from each other. To which extent

can the Laurentide and Greenland ice sheets be considered to be independent from each other considering their close proximity? It seems to me that the whole argument rests on this assumption. Or could Greenland be substituted with Antarctica (or would the size of Antarctica change the dynamics)?

---

## Author Comment (AC1) · 23 Oct 2020

**M. Verbitsky** (verbitskys@gmail.com)

Dear Anonymous Referee #1, Thank you for your review. The following is our response to your two questions.

**Comment:** In your model Eqs. (1) - (5) you represent two ice sheets (Laurentide and Greenland) which are independent from each other. To which extent can the Laurentide and Greenland ice sheets be considered to be independent from each other considering their close proximity? It seems to me that the whole argument rests on this assumption.

**Answer:** The equations (1) – (5) and corresponding diagram of Fig. 1(a) describe the dynamical system when the Greenland ice sheet is coupled one-way via the climate temperature ω acting as an external forcing for it. We used this configuration for our scaling reasoning. For the numerical experiment, we added a term $-\gamma_2 S_G$ in the equation (3) thus making full coupling of the Laurentide and Greenland ice sheets (lines 122-124). Full coupling is, indeed, important. As we demonstrated earlier (Verbitsky et al, 2019, https://doi.org/10.5194/esd-10-257-2019), millennial forcing may penetrate upscale and under some conditions may significantly change the orbital-timescale dynamics.

**Action:** To articulate this more clearly, we suggest adding one more panel to Fig. 1 that would show both dynamical configurations:

[Figure]

Fig. 1: (a) the dynamical system (1) – (5) as it has been used for scaling reasoning. The Greenland ice sheet is coupled one-way via the climate temperature ω acting as an external forcing for it. Red circles mark positive feedback loops and green circles mark negative feedback loops; (b) the dynamical system (1) – (5) as it has been used in the numerical experiment. The Laurentide and Greenland ice sheets are fully coupled. Pink circles mark weaker positive feedback loops.

**Comment:** Or could Greenland be substituted with Antarctica (or would the size of Antarctica change the dynamics)?

**Answer:** Your observation regarding the proximity is correct: We do not consider here a number of mechanisms that might be important in the close proximity of Laurentide ice sheet - our coupling is made through the global albedo and global temperature only. Nevertheless, we cannot simply replace the Greenland ice sheet with the Antarctic ice sheet. Even though the area of the Antarctica is also limited, we may anticipate that the larger area of the Antarctic glaciation implies thicker ice sheet, bigger Peclet number, weaker internal negative feedback, larger amplitude and period of the relaxation oscillations, and, therefore, possibly stronger influence on the global climate including Laurentide ice sheet. Besides,

Antarctic ice sheet volume changes cause stronger sea-level variations, which are immediately distributed worldwide and would immediately affect Laurentide ice sheet, not via the global temperature but via sea-level. For that reason, to answer your question confidently, an additional study may be needed.
**Action**: We will add this discussion into the text.

---

## Referee Comment (RC2) · Anonymous Referee #2 · 29 Oct 2020

This manuscript presents a further development of the Verbitsky et al. papers (2018 ; 2019) that try to capture Quaternary climate variability. Following many previous suggestions that the millenial scale climate variability and the Milankovitch one are strongly connected, the authors attempt to provide a common simple framework for these two different modes of climate changes using a conceptual model. This model is more complex than its predecessor, with 5 dynamical variables (instead of 3) and about 16 parameters (instead of about 11) if I am correct. The output of the model is not actually compared to observation, but only shows "more variability" in both the 100-kyr band and the millennial band. In other words, I find that the complexity of the model does not scale reasonably well with its results. But more importantly, I do

not agree with some basic assumptions made by the authors, as explained below: standard knowledge shows that the Peclet number should increase when the ice sheet size decrease, not the opposite. I therefore cannot recommend publication of this manuscript.

Main comments:

1 - The key element of this paper is the Peclet number defined as "a H/k", with "a" the mass balance, "H" the ice thickness and "k" the temperature diffusivity. The authors assume that this number is an increasing function of ice-sheet size. But standard knowledge of ice-sheets suggests that this is very likely to be just the opposite. Indeed, today we have a rather good knowledge of two ice-sheets (Greenland and Antarctica). The size of Greenland is smaller than the size of Antarctica and its height H is roughly 70% or 80% of the one of Antarctica. But the advection parameter "a" (ice accumulation minus ablation) is certainly much higher in Greenland (about 5 to 10 times higher). This is partly due to atmospheric circulation and continental set-up, but mostly due to ice sheet height: the larger the ice sheet, the higher its surface, the drier the climate. I therefore do not think reasonable to assume that the Peclet number defined above would increase with ice sheet size. On the contrary, I expect it to decrease strongly. Overall, diffusion should dominate the dynamics of large ice sheets, since they have very little precipitations. This is likely the case for the Laurentide ice sheet in the past, as it is for Antarctica today. I therefore strongly disagree with the main message of this manuscript.

2 - I also find it difficult to appreciate the relevance of such a model without any result in the time domain. If I understand well the wavelet diagram, the model exhibits a very strong 400-ky oscillation that is certainly not observed (the famous 400-kyr problem. . .). I therefore doubt that this (rather complex) model can bring any insight in the problem of Quaternary climate variability.

---

## Author Comment (AC2)

**M. Verbitsky (verbitskys@gmail.com)**

Dear Anonymous Referee #2, Thank you for your review. The following is our response to your comments.

**Main comments**

**Comment:** The key element of this paper is the Peclet number defined as "aH/k", with "a" the mass balance, "H" the ice thickness and "k" the temperature diffusivity. The authors assume that this number is an increasing function of ice-sheet size. But standard knowledge of ice-sheets suggests that this is very likely to be just the opposite. Indeed, today we have a rather good knowledge of two ice-sheets (Greenland and Antarctica). The size of Greenland is smaller than the size of Antarctica and its height H is roughly 70% or 80% of the one of Antarctica. But the advection parameter "a" (ice accumulation minus ablation) is certainly much higher in Greenland (about 5 to 10 times higher). This is partly due to atmospheric circulation and continental set-up, but mostly due to ice sheet height: the larger the ice sheet, the higher its surface, the drier the climate. I therefore do not think reasonable to assume that the Peclet number defined above would increase with ice sheet size. On the contrary, I expect it to decrease strongly. Overall, diffusion should dominate the dynamics of large ice sheets, since they have very little precipitations. This is likely the case for the Laurentide ice sheet in the past, as it is for Antarctica today. I therefore strongly disagree with the main message of this manuscript.

**Answer:** Let us first examine the claim that "Overall, diffusion should dominate the dynamics of large ice sheets, since they have very little precipitations". The established theory for ice sheets (Grigoryan et al, 1976) concludes that above the basal diffusive boundary layer "the temperature propagates along the particle trajectories... and conductive heat transfer causes small variations in the temperature field". Indeed, an ice sheet is considered here in a "thin-layer" approximation, i.e., it is a glacial object with a vertical dimension that is much smaller than the horizontal dimension. Therefore, horizontal diffusion of heat can be neglected, and the relevant length scale to be included in the Peclet number is the vertical length scale. Now, let us examine the Peclet number. For ice thickness  $H \sim 3,000$  m, ice temperature diffusivity  $k = 10^{-6}$  m2/s, and  $\hat{a} \sim 3x10^{-9}$  m/s (the latter, 10 sm/yr, would correspond to the dry conditions of, for example, East Antarctic ice sheet), the Peclet number  $Pe = \hat{a}H/k \sim 10$ . It means that temperature advection dominates vertical temperature diffusion even for very dry conditions. Extremely low temperature diffusivity of ice is responsible for this phenomenon. We therefore conclude that this first claim of the reviewer is incorrect.

Let us now consider the second claim that the Peclet number would "decrease strongly" for growing ice sheets because ice accumulation is smaller. What should be observed here is that over a full glaciation phase, starting from no ice sheet to a fully mature ice sheet before the deglaciation, the ice sheet undergoes more or less rapid phases of expansion. For example, the final growth phase between marine isotopic stages 3 and 2 was pretty rapid, indicating substantial accumulation. The reason for this is that the mass balance at a given time results from the interplay of multiple feedbacks, and the quoted by the reviewer "continental feedback", causing less precipitation as the ice sheet grow, is only one of them. If, for the sake of the argument, we consider that the accumulation has been roughly constant throughout the growth phase, then we can reasonably conclude that the Peclet number increased along with the growth of the ice sheet. In other words, the Peclet number of *a same ice sheet growing over time* under constant positive mass balance increases (as we say in the paper "*the Peclet number, changing in concert with the glaciation size*"). The main premise of the paper - The Laurentide ice sheet has the space to grow its Peclet number over time, but the Greenland ice sheet does not - is based on this property. We thus conclude that the second claim of the reviewer is incorrect.

Comparing the Antarctic and Greenland ice sheets is partly misleading for two reasons: first this is a specific instant in the glacial-interglacial cycle (precisely when these ice sheets no longer grow), which is not representative of the dynamics of ice sheet formation and growth. Second, it compares *different ice sheets of different geographical locations*. Nevertheless, the reviewer argument is helpful to us because it lead us to identify an inaccuracy in lines 81-83: When we compare Greenland and hypothetical "big ice sheet", we stated that Greenland's Peclet number may be smaller because the ice sheet is thinner. In fact we tacitly assumed that the mass balance of these two ice sheets is similar. In the new version we will clarify this point.

Action: We will include some elements of the above discussion in the text to make our thinking more explicit.

**Comment:** I also find it difficult to appreciate the relevance of such a model without any result in the time domain. If I understand well the wavelet diagram, the model exhibits a very strong 400-ky oscillation that is certainly not observed (the famous 400-kyr problem...). I therefore doubt that this (rather complex) model can bring any insight in the problem of Quaternary climate variability.

**Answer:** First, we would like to reassure the reviewer about the absence of spurious 400-ka periodicity in the VCV18 (Verbitsky et al. 2018) model. The original publication (Verbitsky et al. 2018) shows side by side continuous wavelet spectra of the LR04 benthic stack and of model output (Figure 13), and the slight yellow band at 400 ka is present in both the record and the model output, suggesting at least that the model passes the spectral test. In fact, that article discusses how 400-ka periodicity could be generated by further altering the balance between positive and negative feedbacks in this model (the so-called *V*-number). Hence, we would consider that, in the context of this note, this otherwise interesting subject does not deserve more discussion.

Our choice of a continuous wavelet representation has been guided by its convenience in allowing visualization of both orbital and millennial variations on the same diagram. Given the space restriction of the "idea" ESD format we propose to maintain this approach. However, for the reference to the reviewer (and interested reader) and for the sake of transparency, the time series of the Greenland area of glaciation that has been used as a source of the wavelet diagram is provided here (Fig. AC2-1).

Fig. AC2-1: Simulated 3-My time-series of the Greenland ice sheet area  $S(10^6 \text{ km}^2)$

Action: We do not suggest any actions here.

**Other comments**

**Comment:** The output of the model is not actually compared to observation, but only shows "more variability" in both the 100-kyr band and the millennial band. In other words, I find that the complexity of the model does not scale reasonably well with its results.

**Answer:** The purpose of our paper is to provide novel explanation to the observed coherence between millennial and orbital-scale variabilities: specifically, after the MPT, millennial events are more frequent and more intense than before. Therefore, our focus on the model output that demonstrates such coherence should not be surprising.

The success of a theory is indeed in good part relates to how much it explains given a small number of adjustable parameters (e.g. Hitchcock and Sober, 2004). This is in fact the main selling point of our VCV18 model which, in our view, is a step forward on previous low order models of glacial-interglacial cycles. We have demonstrated (Verbitsky and Crucifix, 2020) that our model has a property of similarity and a number of parameters can be reduced to only two adimensional parameters. Therefore we have the following arguments to bring here:

- Our reasoning remains heavily dependent on physical constraints and established principles of glaciology. This does not leave so much room for playing around with parameters in a way that would force the conclusions that we want to see

- We are here in the format of an "idea" paper: our objective must be to frame a hypothesis on the basis of established theoretical elements and observations. We certainly agree that this "idea" requires further investigation and deserves being challenged, but the question here is whether we have provided enough arguments to make it credible; that is, whether this augmented VCV18 model provides a formal framework which is credible to make the hypothesis plausible. We definitely believe that this is the case.

Action: We welcome suggestions of the editor if further clarifications about the epistemological status of our hypothesis are needed.

**Reference**

Grigoryan, S. S., Krass, M. S., and Shumskiy, P. A.: Mathematical model of a three-dimensional nonisothermal glacier, Journal of Glaciology, 17, 77, 401-418, 1976

Hitchcock C. and E. Sober: Prediction Versus Accommodation and the Risk of Overfitting, Br J Philos Sci, 55, 1-34, doi:10.1093/bjps/55.1.1, 2004

Verbitsky, M. Y. and Crucifix, M.: π-theorem generalization of the ice-age theory, Earth Syst. Dynam., 11, 281–289, https://doi.org/10.5194/esd-11-281-2020, 2020

Verbitsky, M. Y., Crucifix, M., and Volobuev, D. M.: A theory of Pleistocene glacial rhythmicity, Earth Syst. Dynam., 9, 1025-1043, https://doi.org/10.5194/esd-9-1025-2018, 2018